# ROI-Guided Point Cloud Geometry Compression Towards Human and Machine Vision

## ABSTRACT

Point cloud data is pivotal in applications like autonomous driving, virtual reality, and robotics. However, its substantial volume poses significant challenges in storage and transmission. In order to obtain a high compression ratio, crucial semantic details usually confront severe damage, leading to difficulties in guaranteeing the accuracy of downstream tasks. To tackle this problem, we are the first to introduce a novel Region of Interest (ROI)-guided Point Cloud Geometry Compression (RPCGC) method for human and machine vision. Our framework employs a dual-branch parallel structure, where the base layer encodes and decodes a simplified version of the point cloud, and the enhancement layer refines this by focusing on geometry details. Furthermore, the residual information of the enhancement layer undergoes refinement through an ROI prediction network. This network generates mask information, which is then incorporated into the residuals, serving as a strong supervision signal. Additionally, we intricately apply these mask details in the Rate-Distortion (RD) optimization process, with each point weighted in the distortion calculation. Our loss function includes RD loss and detection loss to better guide point cloud encoding for the machine. Experiment results demonstrate that RPCGC achieves exceptional compression performance and better detection accuracy (10% gain) than some learning-based compression methods at high bitrates in ScanNet and SUN RGB-D datasets.

## CCS CONCEPTS

• **Theory of computation → Data compression**.

## KEYWORDS

Point Cloud Geometry Compression, Human and Machine Vision.

## 1 INTRODUCTION

With the advancement of multimedia technologies, immersive multimedia experiences increasingly capture public attention. Point clouds, epitomizing three-dimensional (3D) representation, offer exact depictions of objects and scenes. Their extensive application, like virtual reality, augmented reality, and autonomous driving, underscores their significance. Unlike two-dimensional (2D) images, point clouds offer a more visceral and comprehensive sensory experience. A point cloud, quite literally, is a "cloud" formed by an aggregation of points. In computational terms, these are typically stored as a series of Cartesian coordinates, constituting point clouds' fundamental geometry data. However, relying solely on this information sometimes fails to fulfill users' visual requirements. Therefore, additional attributes are often appended to point clouds. These include but are not limited to, the color, normal vectors, *etc*.

As 3D scanning sensors evolve, generating vast amounts of point cloud data, the compression of point clouds emerges as a pivotal process. This compression significantly reduces the size of point cloud data and minimizes storage and transmission pressure. The Moving Picture Experts Group (MPEG) is a frontrunner in point cloud compression. They pioneered the establishment of compressing point clouds' geometry and attributes. Meanwhile, the MPEG unveils two point cloud compression standards. The first, the Video-based Point Cloud Compression (V-PCC) [50] employs established video encoding techniques to compress the geometry and attribute of point cloud sequences. The second is the Geometry-based Point Cloud Compression (G-PCC) [25], which encodes geometry using an octree or a trisoup model. Based on the compressed geometry, attribute information can also be encoded either lossy or losslessly.

As neural network-based solutions achieve significant success in image and video compression, learning-based technology gradually finds its application in point cloud compression. Numerous end-to-end point cloud compression methods are proposed. Such as point-based compression methods [9, 14, 44, 47, 48]. Most of these methods take the point cloud as input, use networks like PointNet [31] or PointNet++ [32] for feature extraction, and then construct a variational autoencoder for point cloud compression. However, these methods are mostly limited to handling a fixed number of points, and their compression performance is constrained. Lately, there are many point cloud compression methods based on octrees, such as [8, 13, 35, 37]. In these methods, each octree node is represented by an eight-bit occupancy code. The occupancy codes are then probabilistically estimated using neural network architectures such as Multilayer Perceptron (MLP) or Transformer [6]. Meanwhile, numerous point cloud compression strategies is surrounded by sparse tensors, as in [16, 26, 27, 41, 42]. These methods convert the point cloud into a unified format that intertwines coordinates with attribute features, subsequently applying multi-scale sparse convolution and Arithmetic Encoder to code the point cloud.

Despite numerous research efforts in point cloud compression, they primarily focus on fidelity optimization. However, in practical applications, the purpose of compression is to facilitate machine analysis. For instance, in autonomous driving, lossy compression significantly loses detailed information on the outdoor scenes. In 3D reconstruction, the compression process can also result in the loss of semantic information (Contour) in scene data. Therefore, we propose a novel compression paradigm for scene point clouds, drawing inspiration from [2, 24, 28]. This paradigm can enhance point cloud detection performance during compression. A critical characteristic of object detection is the accurate localization of the bounding box in the point cloud, which is why our approach strongly emphasizes the coding of the ROI region. We establish a geometry compression

framework for point clouds, including base and enhancement layers. The base layer (BL) is responsible for encoding geometry information. The enhancement layer (EL) adopts weighted supervision, guided by masks from the ROI prediction network (RPN) and ROI Searching Network (RSN). Additionally, we weight the distortion function (RD) calculation in a per-point manner according to the generated mask values. To further enhance the effectiveness of the coding process, we incorporate detection loss into the compression task, enabling joint optimization of compression and detection. In summary, the main contributions include:

- To address the issue of semantic information loss in point cloud compression, we propose an ROI-guided point cloud compression paradigm, which optimizes both machine perception performance and visual fidelity simultaneously.
- We divide the point cloud encoding into base and enhancement layers, with the former encoding coordinates and the latter encoding residual information. The mask information from our RPN and RSN supervises these residuals. We also incorporate mask information into the distance calculation of RD optimization in a point-wise manner, aiding in focusing on critical areas during encoding. Additionally, our loss function integrates detection loss to more effectively guide the point cloud encoding process tailored for detection tasks.
- We develop a Multi-Scale Feature Extraction Module (MS-FEM) to extract semantic features. This module captures the local details of the point cloud in efficient manner. In addition, we design a Semantic-aware Attention Module (SAM) to enhance the extraction of semantic information during the encoding and decoding transformation.
- Experimental results show that our method demonstrates excellent compression performance on ScanNet and SUN RGB-D test datasets and achieves more accurate point cloud detection results at high bitrates than traditional and learning-based compression methods.

## 2 RELATED WORKS

### 2.1 Image Compression for Machine

The progression of deep learning has resulted in a surge in utilizing encoded media, such as images, videos, and point clouds. The recent emphasis has shifted towards coding techniques optimized for machine vision in image and video compression. For example, Bai *et al.* [1] introduce a cloud-based, end-to-end image compression and classification model using modified Vision Transformers (ViT). Liu *et al.* [19] propose a scalable image compression method for machine and human vision, featuring a pyramid representation for machine tasks. Yang *et al.* [49] review Video Coding for Machine (VCM) framework, which integrates feature-assisted coding, scalable coding, and intermediate feature coding, *etc*. Meanwhile, there are many approaches optimized for ROI areas in image compression. For example, Cai *et al.* [2] introduce an image compression scheme with ROI encoders/decoders, featuring multi-scale representations, an implicit ROI mask, and a soft-to-hard ROI prediction scheme for effective optimization. Prakash *et al.* [29] introduce a saliency-based compression model that encodes important image regions at higher bitrates and less significant areas at lower bitrates. The methods [17, 24, 51] also concern the ROI areas coding.

## 2.2 Point Cloud Compression for Human

Existing point cloud compression algorithms focus on fidelity optimization, which encompass data structure representation, encoding/decoding transformations, and entropy estimation. For instance, Huang *et al.* [14] introduce a learning-based point cloud compression method using an autoencoder and sparse coding structure, achieving high compression with minimal loss. Yan *et al.* [48] present an autoencoder-based architecture with a PointNet-based encoder and a nonlinear transformation decoder for lossy geometry point cloud compression. Quach *et al.* [33] integrate 3D convolution neural network (CNN) into an encoder-decoder transformation. Huang *et al.* [13] introduce a compression algorithm for LiDAR point clouds, utilizing octree encoding and a tree-structured conditional entropy model to exploit sparsity and structural redundancy. Que *et al.* [35] propose a two-stage learning-based framework for static and dynamic point cloud compression, combining octree and voxel representation for context estimation. Wang *et al.* [43] divide point clouds into distinct blocks, predicts context using super-priority, and utilizes a arithmetic encoder for compression. Wang *et al.* [41] also introduce a unified point cloud geometry compression using multiscale sparse tensor-based voxelization. Although researchers have made many attempts at point cloud compression, they have yet to consider the downstream task performance.

### 2.3 Point Cloud Compression for Machine

Xie *et al.* [46] introduce a coding network utilizing sparse convolution and design to extract semantic information for classification tasks concurrently. Ulhaq *et al.* [40] present a PointNet based codec specialized for classification, offering a superior rate-accuracy trade-off with significant BD-Rate reduction and propose two lightweight configurations for low-resource devices. Ma *et al.* [23] propose the human-machine balanced compression method, which utilizes a pre-trained lightweight backbone and a semantic mining module for multi-task feature aggregation, balancing signal and semantic distortion with multi-task learning. Liu *et al.* [18] propose a learning-based point cloud compression framework optimized for object detection, featuring a gradient bridge function for seamless codec-detector integration and a progressive training strategy, achieving significant compression with maintained detection accuracy. Liu *et al.* [20] introduce a new point cloud compression framework for human and machine vision, featuring a two-branch structure with a shared octree-based module and a point cloud selection module for sparse point optimization. Although the above methods involve machine perception, most of them are simplistic and do not achieve joint optimization or supervision.

## 3 METHODOLOGY

### 3.1 Overall Framework

We propose a point cloud geometry compression scheme supervised by the ROI region, designed to simultaneously optimize the fidelity and the detection performance of point clouds. As illustrated in Fig. 1, our approach comprises the following steps:

**Encoding and Residual Generation.** Given the input point cloud $(x_0, y_0, z_0)$, the process begins with quantization ($Q$). Subsequently, the quantized point cloud follows two paths. The first path involves G-PCC encoding to generate a coarse geometry bitstream,

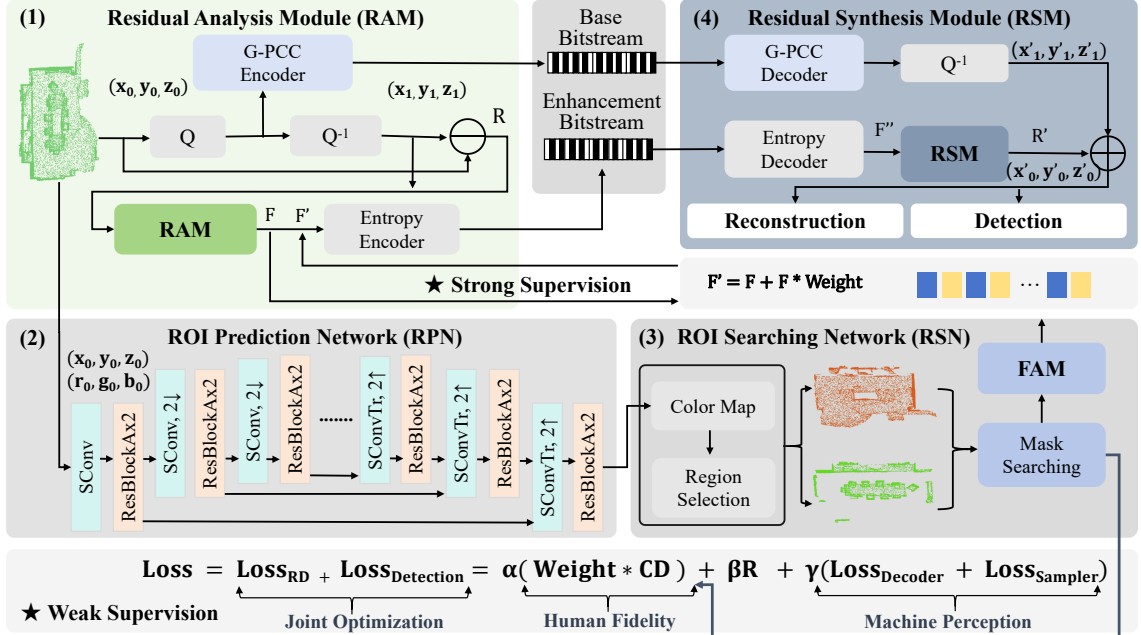

**Figure 1: The overview of our proposed RPCGC framework includes: (1) The Residual Analysis Module (RAM) shows the coordinate coarse coding process. (2) The ROI prediction network (RPN) is the probability prediction network. (3) The ROI Searching Network (RSN) involves processing the predicted mask and applying it as a weight to the residual features. FAM denotes Feature Alignment Module. Residual Synthesis Module (RSM) denotes the decoding process. When the original point cloud is reconstructed, we then feed the reconstructed data into the detection network for further analysis. $Q$ and $Q^{-1}$ mean quantization and de-quantization. $\ominus$ and $\oplus$ denote tensor element-wise subtraction and addition operation.**

forming the BL bitstream. The second path involves de-quantizing the quantized point cloud using $Q^{-1}$ to retrieve the geometry coordinates $(x_1, y_1, z_1)$. The residual information $R$ is calculated by subtracting these obtained coordinates from the original input. Meanwhile, the RAM extracts finer-grained features $F$ from $R$.

**ROI Prediction and Mask Generation.** Concurrently, the original input point cloud undergoes processing via our specially designed RPN to create masks. This RPN employs a sparse convolutional U-Net architecture [36], accepting color or instance labels for each point and yielding probability outputs.

**Mask Processing and Feature Enhancement.** The instance information predicted by the RPN is converted into a 3D mask through the RSN. This mask is weighted into feature $F$, forming a new semantic-information-bearing $F'$. Then, the $F'$ is encoded into an EL bitstream by an Entropy Encoder. Moreover, the mask influences the RD optimization process, which guides the bitrate allocation towards the target regions within the point cloud.

**Decoding and Analysis.** On the decoding side, the coarse point cloud from the BL is decoded using a G-PCC decoder, and dequantization $(Q^{-1})$ yields $(x'_1, y'_1, z'_1)$. An Entropy Decoder decodes the EL bitstream to obtain $F''$, which, through an RSM, reconstructs the residual features $R'$. The restored point cloud $(x'_0, y'_0, z'_0)$ is obtained by summing $R'$ with $(x'_1, y'_1, z'_1)$. Then, $(x'_0, y'_0, z'_0)$ is put into the Group-Free [21] for detection analysis. Furthermore, the loss function from the detection is integrated into the RD optimization.

## 3.2 Formulation

**Residual Weighting.** Our research introduces a hierarchical mask mechanism that significantly boosts performance. We achieve this

enhancement by applying weights to both the residual feature and the loss function. Assuming the input point cloud is $x$, we process it through a RPN. Inspired by the U-Net [36] architecture, our RPN adopts a similar approach to U-Net's encoding and decoding stages, enabling more effective feature reuse. The network's encoding path of RPN captures essential low-level features, such as coordinates and local properties, while the decoding path reconstructs higher-level semantic features. Subsequently, the output $(x_{roi})$ of RPN is fed into RSN. Adopting the region searching algorithm, we divide the original point cloud into foreground color map $x_{fg}$ and background color map $x_{bg}$. We calculate the nearest neighbor distance of the residual feature map $x_{enh}$ and search the corresponding mask value for each element in the residual feature map (Nearest neighbor matching process). We obtain a new mask $x_m$, and the residual feature map is weighted by $x_m$ through the above process. We describe the process as below:

$$
\begin{aligned}
x_{roi} &= Max(RPN(x)), \\
\{x_{fg}, x_{bg}\} &= RSN(x_{roi}), \\
x_{cm} &= L2\_Norm(\{x_{fg}, x_{bg}\}), \\
x_m &= Conv(ReLU(Fc(x_{rm}))), \\
x'_{enh} &= x_{enh} * (1 + x_m),
\end{aligned}
\tag{1}
$$

where $RPN(\cdot)$ and $RSN(\cdot)$ refer to the RPN and RSN, respectively. $L2\_Norm(\cdot)$ represents the nearest neighbor search algorithm. $Fc(\cdot)$, $ReLU(\cdot)$, and $Conv(\cdot)$ denote the Fully Connected layer, Rectified Linear Unit, and Convolution operation, respectively. $x'_{res}$ is the weighted feature map in the EL.

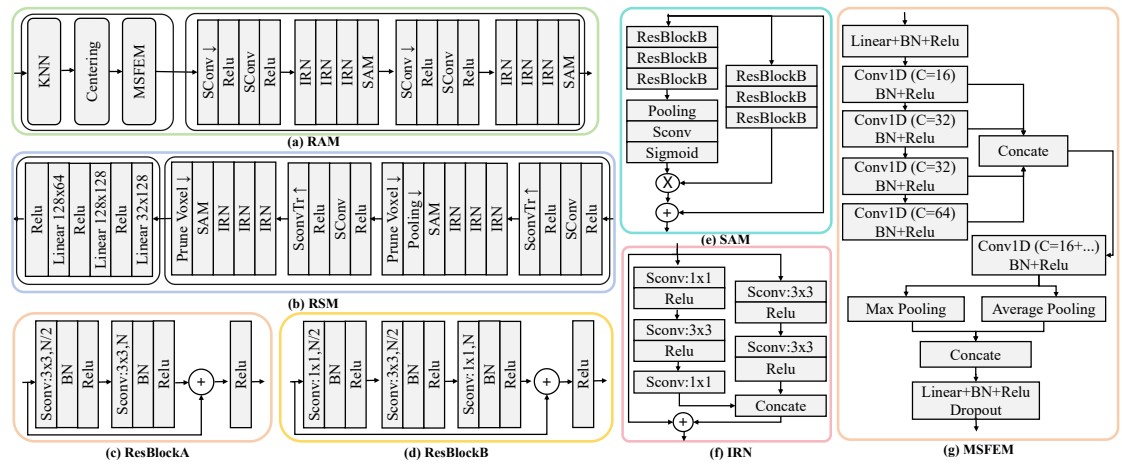

**Figure 2: The overview of the network in RPCGC. (a) "RAM" stands for the Residual Analysis Module, (b) "RSM" denotes the Residual Synthesis Module. (c) "ResBlockA" refers to the residual structure employed in RPN. (d) "ResBlockB" describes the residual structure in SAM [3]. (e) "SAM" represents the Semantic-aware Attention Module. (f) "IRN" signifies the Inception-Residual Network. (g) "MSFEM" stands for the Multi-Scale Feature Extraction Module. All the convolutions in the network are 3D sparse convolutions. $N$ represents the dimension of the input point cloud, and $C$ denotes the number of channels.**

**Loss Weighting.** We utilize Information Bottleneck (IB) theory [39] to optimize the loss function. For the human vision task $v$ and machine vision task $t$, we assume $I(\cdot)$ denotes mutual information function. $p(v, t \mid x)$ represents the mapping process from the input point cloud to the latent representation, and then to the machine tasks and the reconstructed point cloud. $\mu$ and $\tau$ denote the Lagrange multiplier of human and machine vision. Since $x_{base}$ and $x'_{enh}$ fully conditioned on $x$ are completely dependent, then:

$$\min_{p(v,t|x)} I(x; x_{base}, x'_{enh}) - \mu I(x_{base}, x'_{enh}; \overline{x})$$
$$- \tau I(x_{base}, x'_{enh}; t), \tag{2}$$

$$I\left(x; x_{base}, x'_{enh}\right) = H\left(x_{base}, x'_{enh}\right) - H((x_{base}, x'_{enh}) \mid x),$$
$$= H(x_{base}) + H(x'_{enh}), \tag{3}$$

where $H(\cdot)$ and $H(\cdot \mid \cdot)$ denote the entropy and the conditional entropy, respectively. Therefore, we can obtain:

$$\min_{g_a, g_s, g_p, g_m} H(x_{base}) + H(x'_{enh}) + \lambda_h D_h + \lambda_m D_m, \tag{4}$$

where $g_a$, $g_s$, $g_p$, and $g_m$ denote the transformation of RAM, RSM, RPN, and RSN, respectively. $\lambda_h$ and $\lambda_m$ represent the weight of human and machine vision distortion. We support $\lambda_h D_h$ as the proxy for $\mu I\left(x_{base}, x'_{enh}; \overline{x}\right)$, and $\lambda_m D_m$ as a surrogate for $\tau I\left(x_{base}, x'_{enh}; t\right)$. Moreover, we apply the weight mask $x_m$ to the Chamfer Distance (CD) [7], the process can be formulate as:

$$D_{RW-CD}(P_1, P_2) = \frac{1}{P_1} \sum_{a \in P_1} \frac{M_1(a, b)}{\sum M_1(b)} (\min_{b \in P_2} \|a - b\|_2^2)$$
$$+ \frac{1}{P_2} \sum_{b \in P_2} \frac{M_2(a, b)}{\sum M_2(a)} (\min_{a \in P_1} \|b - a\|_2^2), \tag{5}$$

where $P_1$ and $P_2$ represent the two point clouds to be compared. $a$ and $b$ represent the point in $P_1$ and $P_2$, respectively. The RSN generates mask information (probability) for each point, represented by $M_1(\cdot)$. The first term of Eq. (5) incorporates this mask information into the distance calculation between points $(a, b)$. $M_2(\cdot)$

denotes using the nearest neighbor search algorithm to calculate the distance between points $(a, b)$, recording the distance's index. This index allows the retrieval of the closest mask value from $M_1(\cdot)$ to assign to the corresponding point. The second term of Eq. (5) applies weighting the distance calculation between points $(b, a)$ in this manner. Finally, we obtain the total loss function as:

$$\mathcal{L}_{total} = \mathcal{L}_r + \mathcal{L}_v + \mathcal{L}_t,$$
$$= \delta R_{base} + \beta R_{enh} + \alpha D_{RW-CD} + \gamma (D_{sp} + D_{det}). \tag{6}$$

In Eq. (6), $\mathcal{L}_r$, $\mathcal{L}_v$, and $\mathcal{L}_t$ represent rate, distortion, and machine task loss, respectively. $\delta$ and $\beta$ are hyper-parameters of base and enhancement rate loss. While $\alpha$ and $\gamma$ adjust the ROI Weight-CD (RW-CD) and machine task loss. The calculation of $D_{sp}$ and $D_{det}$ is described in Group-Free [21].

## 3.3 ROI Generation and Refinement

Our ROI region process is divided into two procedure: the RPN and the RSN structure, as shown in Fig. 1 (2) and (3).

**The ROI Prediction Network** employs a U-Net architecture built by the Minkowski Engine [4]. Its foundational block in the RPN, integrates a Sparse Convolution (SConv) with two Residual Block A (ResBlockA) layers [12] to create a feature extraction module. This module conducts down-sampling through a combination of SConv and ResBlockA, and up-sampling by pairing Sparse Transposed Convolution (SConvTr) with ResBlockA, incorporating a skip connection operation. These skip connections effectively link layers of matching stride sizes, ensuring smooth feature transitions between the down-sampling and up-sampling phases. The structure of ResBlockA, shown in Fig. 2 (c), facilitates fine-grained feature extraction through residual connections. The input point cloud labeled with instance tags, and the RPN outputs the probability of the predicted label. Meanwhile, the RPN utilizes Cross Entropy as the loss function for optimization.

**The ROI Search Network** includes multiple steps, such as Color Map, Region Selection, and Mask Searching. In the Color Map, the output probabilities of the RPN are converted into category labels

using Softmax. During the Region Selection, background and foreground are identified, "furniture", "doors", "floors", and "walls" are selected as the background, with the remaining categories comprising the foreground. The Mask Searching process transforms the foreground and background into a mask tensor. For the mask information of the foreground area, we apply it to the residual features with double the weight. For the background area, we maintain the original mask values unchanged. Then, the Feature Alignment Module (FAM), which includes an FC layer and a convolution layer, aligns the generated mask to match the dimensions of the residual feature. The aligned result is then multiplied by the residual feature map, as described in Eq. (1). This process is referred to as strong supervision. Meanwhile, we apply weak supervision to the RD loss optimization process through per-point weighting as in Eq. (5).

### 3.4 Encoder and Decoder

**Encoder.** We adopt the residual coding method commonly used in multimedia coding [28]. As shown in Fig. 1 (1), this process following a coarse-to-fine coding strategy. Suppose the input is denoted as $(x_0, y_0, z_0)$, and it first undergoes coordinate quantization, followed by a rounding operation for integer conversion. Then, we employ the G-PCC codec to compress the quantized coordinates, thus generating the BL bitstream. Lately, we utilize the de-quantization operation $Q^{-1}$ to obtain finer geometry coordinates $(x_1, y_1, z_1)$. These reconstructed coordinates are differenced from the original coordinates $(x_0, y_0, z_0)$ to extract residual information $R$.

The next step involves feature extraction and residual encoding using the RAM and Entropy Encoder, as shown in Fig. 2 (a). The RAM is divided into two sub-modules: the first is feature extraction, combining KNN [11], Centering, and MSFEM (Fig. 2 (g)) to extract latent features of the residual. Compared to PointNet, MSFEM is more adept at extracting multi-scale and local details information from the point cloud. The second module comprises two down-sampling stages, which utilize sparse convolution, three Inception-Residual Networks (IRN: Fig. 2 (f)), and one SAM (Fig. 2 (e)), to further refine the features of the residual. Constructed from Residual Blocks B (ResBlockB: Fig. 2 (d)), the SAM expands the receptive field and enhances performance in downstream tasks. Our experimental results highlight the significant role of the SAM in improving the detection performance. Let $x$ be the input tensor, $Conv_{k,s,d}(\cdot)$ a convolution with kernel size $k$, stride $s$, and dilation $d$, $BN(\cdot)$ batch normalization, $ReLU(\cdot)$ the ReLU activation function, $\oplus$ tensor element-wise addition, and $DS(\cdot)$ down-sampling of $x$ if needed. Then, the SAM is defined as:

$$
\begin{aligned}
out_1 &= ReLU(BN(Conv_{1,1,1}(x))), \\
out_2 &= ReLU(BN(Conv_{3,s,d}(out_1))), \\
out_3 &= BN(Conv_{1,1,1}(out_2)), \\
res &= \begin{cases} DS(x) & \text{if downsample,} \\ x & \text{otherwise,} \end{cases} \\
output &= ReLU(out_3 \oplus res),
\end{aligned}
\tag{7}
$$

where $out_1$, $out_2$, and $out_3$ represent the outputs of the consecutive layers in the SAM. The final output is the ReLU activation of the element-wise sum of $out_3$ and the residual (either $x$ or the down-sampled $x$). After extracting features from the residuals twice, we employ an entropy model for encoding, obtaining the BL bitstream.

**Decoder.** As shown in Fig. 1 (4), the decoder involves two main steps. Initially, the G-PCC Decoder decodes the BL bitstream, and then de-quantization follows to produce $(x_1', y_1', z_1')$. Subsequently, an Entropy Decoder decodes the EL bitstream to generate $F''$, which is then input into the RSM for feature up-sampling. The RSM includes two fundamental components: the first is a feature up-sampling layer that uses two SConvTr blocks and "Prune Voxel" plus SAM block, as shown in Fig.2 (b). Meanwhile, the "Prune Voxel" step removes empty voxels for better reconstruction. The second component, a feature restoration module, involves two linear layers that produce the residual $R'$. Finally, the process adds $R'$ to $(x_1', y_1', z_1')$ to reconstruct the point cloud $(x_0', y_0', z_0')$, which is then used for object detection through the Group-Free. Additionally, the detection loss function is added to the RD optimization process.

### 3.5 Optimization

We implement a two-stage optimization process. The first stage encompasses the ROI prediction network, optimized using a Cross Entropy loss function. The second stage focuses on RD optimization, which splits the loss function into two parts. Initially, we address RD optimization for compression tasks by calculating distortion using our unique weighted distortion strategy. Subsequently, we feed the reconstructed point cloud into the detection network to compute the detection loss. This loss becomes an integral part of the overall RD optimization process. Our versatile framework enables joint optimization for various downstream tasks, such as classification and segmentation. The loss function is described as:

$$
\mathcal{L}_{RPN} = \frac{1}{N} \sum_i L_i = -\frac{1}{N} \sum_i \sum_{c=1}^M y_{ic} \log(p_{ic}),
\tag{8}
$$

$$
\mathcal{L}_{\text{detection}} = \frac{1}{L} \sum_{l=1}^L \mathcal{L}_{\text{decoder}}^{(l)} + \mathcal{L}_{\text{sampler}},
\tag{9}
$$

$$
\begin{aligned}
\mathcal{L}_{total} &= \alpha D + \beta R + \gamma \mathcal{L}_{\text{detection}}, \\
&= \alpha D_{\text{RW-CD}} + \beta R + \gamma \mathcal{L}_{\text{detection}}.
\end{aligned}
\tag{10}
$$

In Eq. (8), $M$ is the total number of categories, $c$ specifies a particular category, and $N$ denotes the overall class count. $y_{ic}$ indicates each point cloud's assigned category, and $p_{ic}$ represents the predicted probability. In Eq. (9), the first item of $\mathcal{L}_{detection}$ represents the average loss across all decoding stages of the decoding head in Group-Free, where $L$ signifies the stage of decoding. The loss $\mathcal{L}_{decoder}$ aggregates five distinct types of losses: objectness prediction, box classification, center offset prediction, size classification, and size offset prediction losses. The second item of $\mathcal{L}_{detection}$ represents the loss function of the Group-Free sampling head $\mathcal{L}_{sampler}$. Eq. (10) describes the RD optimization integrated with the detection task, utilizing parameters consistent with those in Eq. (6).

## 4 EXPERIMENTS

### 4.1 Experimental Conditions

**Training Dataset.** We employ two distinct datasets for compression and detection tasks. The first dataset, ScanNetv2 [5], is an indoor scene dataset that includes bounding box labels, as well as semantic and instance segmentation labels for objects. ScanNetv2 consists of 1,513 point clouds spanning 18 categories. The training dataset of ScanNetv2 includes 1,201 point clouds, while the testing

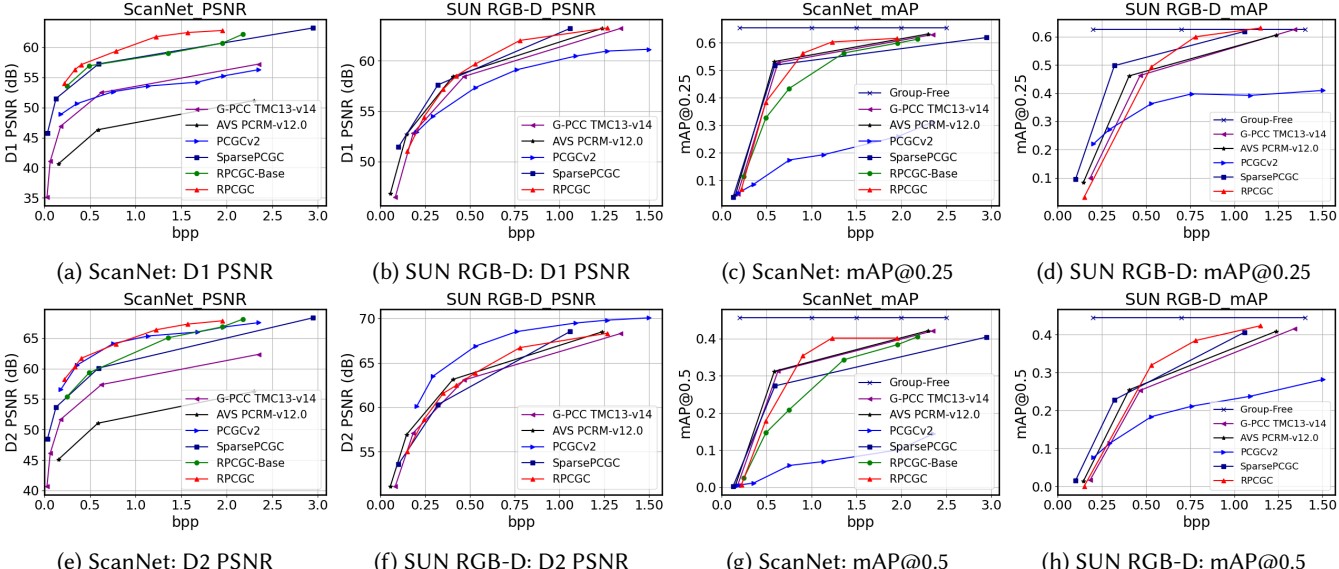

**Figure 3: Performance comparison using Rate-Distortion (RD) and Rate-Detection (R-mAP) curves under different bitrates. (a), (e), (b), and (f) show the RD curves on the ScanNet and SUN RGB-D, and the RPCGC-base represents a scenario where no optimization strategies are added. Meanwhile, (c), (g), (d), and (h) present the R-mAP curves on the ScanNet and SUN RGB-D.**

dataset contains 312. We initially sample each point cloud to 50,000 points and then reduce it to approximately 30,000 points through quantization. The second dataset is SUN RGB-D [38], a single-view indoor dataset comprising 50,000 RGB and depth images. Each associated point cloud includes semantic labels and bounding box information. While the original annotations cover 37 object categories, our detection efforts concentrate on the ten most prevalent categories. We sample the original point clouds of SUN RGB-D to approximately 20,000 points, which serve as inputs for our compression network. The SUN RGB-D test dataset includes 500 point clouds. For brevity in subsequent experiments, we will refer to ScanNetv2 as ScanNet.

**Training Setting.** We utilize the ScanNet and SUN RGB-D datasets to train RPN architecture, adhering to the standard data partitioning strategy outlined in [21, 30]. We employ a Momentum SGD optimizer with an initial learning rate at $1e^{-1}$ throughout the training process of RPN. We incorporate various data augmentation techniques, such as random scaling, rotation, and translation, to enhance the model's generalization capabilities. Following previous research practices, we select the mean Intersection over Union (mIoU) as evaluation metrics for the RPN model to ensure the reliability and consistency of the results.

In the training phase of the compression task, we apply the RD plus detection loss defined in Eq. (10) as our optimization function, and the training parameters are consistent with Group-Free [21]. We set the rate loss parameter $\beta$ to 1, the distortion parameter $\alpha$ to $\{1, 2, ..., 5\}$, the detection parameter $\gamma$ to 0.01, and adjust the quantization value range from 0.15 to 0.45. For measuring distortion in the compression, we utilize D1 PSNR and D2 PSNR [15], with bits per point (bpp) as the rate metric and calculating the gain using DB-PSNR. We employ the following methods as the benchmark: (1) G-PCC [25] TMC13 v22 octree codec and AVS PCRM [10] V12 codec. (2) PCGCv2 [42] and SparsePCGC [41] (abbreviated as SPCGC in

the table) lossy mode without offset prediction. We employ the mean Average Precision (mAP) under different IoU thresholds as the evaluation criterion for the detection task, including mAP@0.25 and mAP@0.5 thresholds. We also consider factors such as the model size and coding runtime. For the detection, we employ the Group-Free as the detection framework for joint training and testing.

It is important to emphasize that due to the relatively low performance of the ROI prediction task (with mIoU only 60%), when calculating the mask values for the SUN RGB-D dataset, we use the original segmentation labels instead of the predictions from the RPN model. This approach is primarily adopted to ensure our weighted method operates on a more accurate prediction base. To evaluate the generalizability of our method on the MPEG dataset, we extend our experiments to the MVUB [22] dataset, which includes the Phil, Sarah, and Queen data samples. These samples are quantized to 10 bits. For a comparative analysis, we incorporate learning-based methods such as pcc_geo_cnn_v1 [33], pcc_geo_cnn_v2 [34], and PCGCv1 [43]. Additionally, we include G-PCC Octree/Trisoup and AVS PCRM codec for a comprehensive evaluation.

## 4.2 Compression Performance

**Human Fidelity Evaluation.** As shown in Fig. 3 (a) and (e), RPCGC demonstrates superior compression performance at a higher bitrate on the ScanNet dataset compared to other learning-based methods. Nonetheless, it still lags slightly behind G-PCC. This discrepancy is primarily attributed to the significant variability in the dataset's distribution, which presents considerable compression challenges for RPCGC. The sampling of the ScanNet dataset consists of 50,000 points, with the quantized data approximately amounting to 20,000 points, and a portion of this distribution is somewhat uneven, placing these data between dense and sparse distributions. Therefore, traditional methods like G-PCC are more effective in restoring the original coordinates, while learning-based methods suffer from a degree of randomness in reconstruction due

Table 1: BD-PSNR (dB) gains measured using both D1 PSNR and D2 PSNR for our proposed RPCGC against the existing methods (anchor) on the ScanNet and SUN RGB-D datasets.

| Anchor | G-PCC[25] | | AVS[10] | | PCGCv2[42] | | SPCGC[41] | |
|---|---|---|---|---|---|---|---|---|
| | D1 | D2 | D1 | D2 | D1 | D2 | D1 | D2 |
| ScanNet | 0.184 | -0.084 | 12.109 | 11.984 | 9.129 | 4.822 | 1.211 | 2.769 |
| SUNRGB-D | 0.838 | 0.576 | 0.252 | -0.185 | 2.834 | -1.414 | -0.126 | 0.308 |
| **Average** | **0.511** | **0.246** | **6.181** | **5.899** | **5.952** | **1.704** | **0.542** | **1.539** |

Table 2: The detection performance comparison using different algorithms on the ScanNet and SUN RGB-D datasets. 0.25 (mAP@0.25) and 0.5 (mAP@0.5) indicate the performance at different thresholds (higher is better).

| Methods | G-PCC[25] | | AVS[10] | | PCGCv2[42] | | SPCGC[41] | | RPCGC | |
|---|---|---|---|---|---|---|---|---|---|---|
| | 0.25 | 0.5 | 0.25 | 0.5 | 0.25 | 0.5 | 0.25 | 0.5 | 0.25 | 0.5 |
| ScanNet | 0.051 | 0.005 | 0.049 | 0.006 | 0.047 | 0.001 | 0.039 | 0.003 | 0.383 | 0.178 |
| | 0.527 | 0.313 | 0.532 | 0.312 | 0.113 | 0.023 | 0.519 | 0.273 | 0.562 | 0.354 |
| | 0.629 | 0.421 | 0.632 | 0.421 | 0.170 | 0.050 | 0.619 | 0.404 | 0.616 | 0.401 |
| SUNRGB-D | 0.099 | 0.017 | 0.085 | 0.015 | 0.221 | 0.077 | 0.095 | 0.016 | 0.032 | 0.001 |
| | 0.463 | 0.253 | 0.461 | 0.257 | 0.363 | 0.184 | 0.498 | 0.228 | 0.600 | 0.384 |
| | 0.623 | 0.416 | 0.606 | 0.409 | 0.392 | 0.238 | 0.619 | 0.406 | 0.631 | 0.423 |
| **Average** | 0.397 | 0.238 | 0.394 | 0.237 | 0.218 | 0.096 | 0.398 | 0.222 | **0.471** | **0.290** |

to the lack of corresponding constraints in the encoding-decoding transformation.

Fig. 3 (b) and (f) show the RD curves of RPCGC against traditional and learning-based approaches in SUN RGB-D. When evaluated using the D1 PSNR, RPCGC exhibits exceptional compression performance at high bitrate but poor performance at low bitrate. Meanwhile, its performance is slightly less than PCGCv2 when assessed using the D2 PSNR. The quantitative analysis shows in Tab. 1 using BD-PSNR metric, reveals that on the ScanNet dataset, RPCGC surpasses G-PCC, AVS, PCGCv2, and SparsePCGC in D1 PSNR by 0.184, 12.109, 9.129, and 1.211, respectively. In D2 PSNR, RPCGC outperforms these models by -0.084, 11.984, 4.822, and 2.769, respectively. On the SUN RGB-D dataset, RPCGC's D1 PSNR gains over the same models are 0.838, 0.252, 2.834, and -0.126, respectively, with D2 PSNR improvements of 0.576, -0.185, -1.414, and 0.308. In summary, RPCGC shows superior compression on ScanNet in high bitrate, indicating enhanced performance over current learning-based methods. However, its efficacy on SUN RGB-D is limited, potentially due to sparser distribution within the dataset.

**Machine Vision Evaluation.** Fig. 3 (c) and (g) display the detection performance of various algorithms on the ScanNet dataset. Unlike other algorithms that put compressed data into a Group-Free detector, RPCGC utilizes a joint optimization approach. This strategy, along with the unique module we designed, enables RPCGC to significantly outperform traditional and learning-based methods at higher bitrate ranges (0.5-3 bpp). However, at lower bitrates, G-PCC and AVS demonstrate superior performance. This is because the reconstructed point clouds at low bitrates lose a significant amount of contour information on the object, leading to a lower detection accuracy of RPCGC. This challenge is consistent with that observed in algorithms like PCGCv2 and SparsePCGC. We

Table 3: The comparison of coding times tested on NVIDIA T4 GPU. "Enc" and "Dec" denote the encoding and decoding time with both units in seconds (lower is better). The coding time of RPCGC is within an acceptable range.

| Methods | G-PCC[25] | | AVS[10] | | PCGCv2[42] | | SPCGC[41] | | RPCGC | |
|---|---|---|---|---|---|---|---|---|---|---|
| | Enc | Dec | Enc | Dec | Enc | Dec | Enc | Dec | Enc | Dec |
| ScanNet | 0.125 | 0.066 | 0.005 | 0.004 | 0.253 | 0.286 | 0.738 | 0.737 | 1.354 | 0.729 |
| SUNRGB-D | 0.081 | 0.055 | 0.001 | 0.001 | 0.248 | 0.269 | 0.589 | 0.586 | 1.243 | 0.707 |
| **Average** | 0.103 | 0.061 | **0.003** | **0.002** | 0.251 | 0.277 | 1.327 | 1.323 | 1.298 | 0.718 |

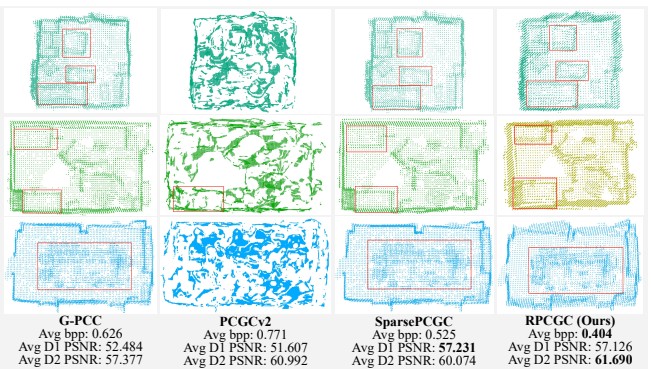

| G-PCC | PCGCv2 | SparsePCGC | RPCGC (Ours) |
|---|---|---|---|
| Avg bpp: 0.626 | Avg bpp: 0.771 | Avg bpp: 0.525 | Avg bpp: **0.404** |
| Avg D1 PSNR: 52.484 | Avg D1 PSNR: 51.607 | Avg D1 PSNR: **57.231** | Avg D1 PSNR: 57.126 |
| Avg D2 PSNR: 57.377 | Avg D2 PSNR: 60.992 | Avg D2 PSNR: 60.074 | Avg D2 PSNR: **61.690** |

Figure 4: The visualization of the detection outputs of different compression algorithms on ScanNet dataset, where the bpp and PSNR represent the average values.

attempt various strategies, including up-sampling, to improve detection performance at low bitrates, yet the improvements remain modest. Furthermore, in practical applications, it is evident that detection performance deteriorates at lower bitrates, reducing its utility, as machine analysis tasks typically operate at higher bitrates. Therefore, our method is likely better suited for applications that require high bitrates specifications.

Fig. 3 (d) and (h) show the rate-detection (R-mAP) curves for the SUN RGB-D dataset. RPCGC exhibits a competitive advantage at higher bitrate ranges. However, it is important to note that the detection performance significantly deteriorates at lower bitrates, which has limited value in practical application. Tab. 2 presents the average detection precision at mAP@0.25 and mAP@0.5 for the ScanNet and SUN RGB-D datasets. Each algorithm computes detection accuracy at a similar bpp. The analysis yields two main conclusions: (1) The detection performance of the point clouds compressed by G-PCC and AVS is consistent. (2) RPCGC surpasses the state-of-the-art detection results of learning-based methods by an average of roughly 10%, highlighting the effectiveness of our designed approach. Further, we evaluated the encoding and decoding times of RPCGC on T4 GPU during inference testing. According to Tab. 3, both the encoding and decoding durations are within an acceptable range. Additionally, our compression model is memory-efficient with a size of 4.5M. Fig. 4 illustrates that our method captures more detailed information than other algorithms.

### 4.3 Ablation Studies

To evaluate the RPCGC's performance, we conducted ablation studies from three perspectives: (1) Variations in the detection network,

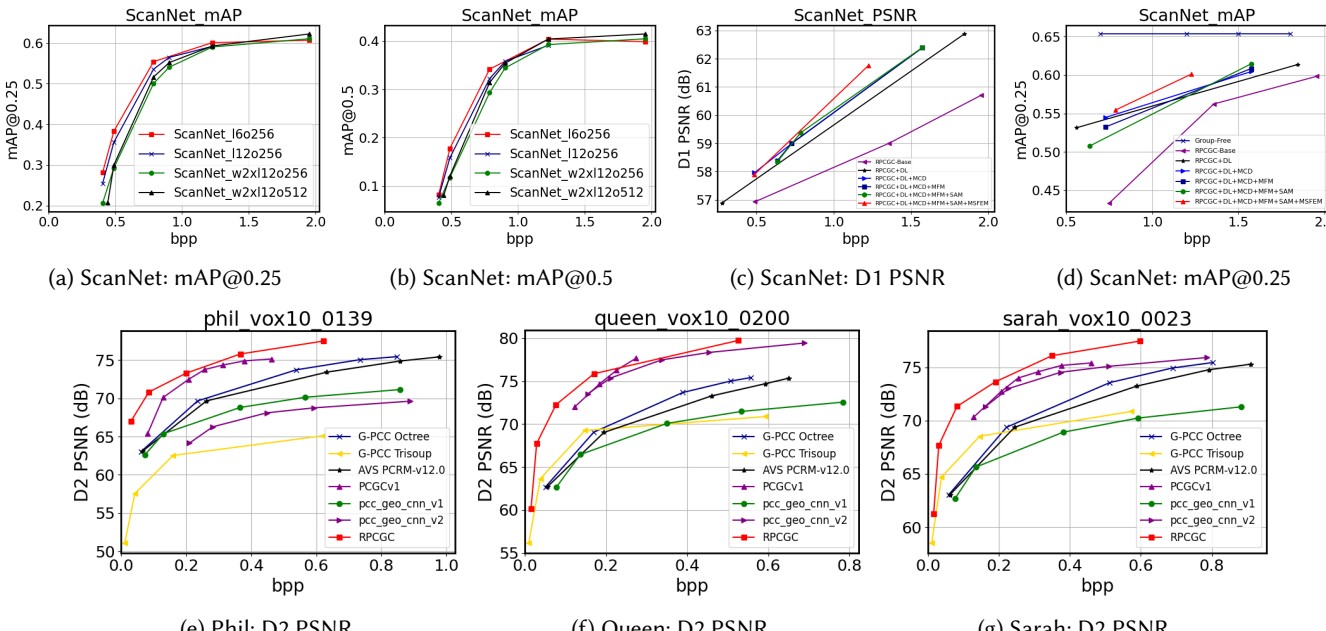

(a) ScanNet: mAP@0.25   (b) ScanNet: mAP@0.5   (c) ScanNet: D1 PSNR   (d) ScanNet: mAP@0.25

(e) Phil: D2 PSNR   (f) Queen: D2 PSNR   (g) Sarah: D2 PSNR

Figure 5: The ablation experiment. (a) and (b) show the detection results of RPCGC on ScanNet using various models in Group-Free. (c) and (d) are different strategies in RPCGC: "DL" denotes Detection Loss, "MCD" means Masking Chamfer Distance Loss, "MFM" signifies masking Residual Map, "SAM" denotes the Semantic-Aware Module, and "MSFEM" means the Multi-Scale Feature Extraction Module. (e-g) show the RD curves of RPCGC and other methods in the MPEG test dataset.

(2) The effects of our designed modules on compression and detection tasks, and (3) The generalizability and robustness of RPCGC across different datasets.

**Detection Models**. We evaluate the effectiveness of our approach through joint testing with different detection models. Specifically, we employ four sets of trained models from Group-Free, which vary in layer level and Transformer depth. As shown in Fig. 5 (a) and (b), the "w2×" signifies that we double the width of the PointNet++ [32] backbone in Group-Free. "L" represents the depth of the decoder, and "O" denotes the number of object candidates. For instance, "(L6, O256)" refers to one with a 6-layer decoder (*i.e.*, six attention modules) and 256 object candidates. Our experiments reveal that, despite using models of varying depths for detection, the differences in detection results were not significant, indicating that the performance of detection tasks is greatly affected by compression distortion, especially at lower bitrates where the detection performance is generally poorer.

**Different Strategies**. In the second experiment, we examine each designed module and strategy to assess their effects on compression and detection tasks. As illustrated in Fig. 5 (c) and (d), RPCGC-Base serves as the baseline model. At the same time, DL, MCD, MFM, SAM, and MSFEM correspond to configurations trained with detection loss, mask-weighted distortion, mask-weighted residual features, a semantic-aware attention module, and a multi-scale feature extraction module, respectively. Fig. 5 (c) indicates that the SAM and MSFEM modules significantly enhance the compression performance. Meanwhile, Fig. 5 (d) reveals that the joint loss from detection and compression tasks markedly enhances detection performance, with the SAM effectively guiding the feature extraction process towards detection-oriented tasks.

**Generalization Test**. In the third experiment, we isolate the designed loss function and ROI prediction network from RPCGC to evaluate their compression performance on MPEG datasets. We trained on the ModelNet [45] dataset and then evaluate the impact of these configurations on the MPEG standard compression dataset. As shown in Fig. 5 (e-g), we plot RD curves based on D2 PSNR indicators on the Phil, Queen, and Sarah, demonstrating that RPCGC surpasses traditional algorithms like AVS PCRM and G-PCC (Octree/Trisoup) codec on specific 10-bit datasets. Additionally, our approach exhibits advantages over learning-based methods, such as pcc_geo_cn_v1, pcc_geo_cn_v2, and PCGCv1.

## 5 CONCLUSION

This paper introduces an innovative point cloud compression method comprising two key components: a BL and an EL. The BL is responsible for processing the geometry coordinates, while the EL focuses on encoding additional residual information of the point cloud. This information is effectively processed and optimized through the ROI prediction network and mask weighting process. We further develop a new RD optimization strategy that incorporates mask information, enhancing the encoding quality of critical areas and integrating detection loss into the total loss function to better guide detection tasks. To extract point cloud contour features more accurately, we design a MSFEM and a SAM in the residual encoding process for finer semantic information extraction. Experiment results demonstrate that our method improves compression performance on the ScanNet and SUN RGB-D datasets and achieves more accuracy of point cloud detection at high bitrates compared to traditional and learning-based point cloud compression methods. In future work, we aim to investigate multimodal point cloud compression alongside analysis tasks in the LiDAR dataset.

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
