# OpenReview forum: "ROI-Guided Point Cloud Geometry Compression Towards Human and Machine Vision"
_acmmm.org/ACMMM/2024/Conference — MM2024 Poster_

### Official Review · Reviewer_BLbN · 2024-05-09

**Rating:** 4
**Confidence:** 1

**Summary:**

The main concept of this paper is the introduction of a novel Region of Interest (ROI)-guided Point Cloud Geometry Compression method for human and machine vision. The paper addresses the challenges of point cloud data compression, focusing on optimizing both machine perception performance and visual fidelity simultaneously. The proposed method involves a dual-branch parallel structure with a base layer encoding geometry information and an enhancement layer refining this by focusing on geometry details. Additionally, the residual information of the enhancement layer undergoes refinement through an ROI prediction network, generating mask information that is incorporated into the residuals. The paper also introduces a Multi-Scale Feature Extraction Module (MSFEM) and a Semantic-aware Attention Module (SAM) to enhance the extraction of semantic information during the encoding and decoding transformation. The method aims to achieve exceptional compression performance and better detection accuracy compared to traditional and learning-based compression methods at high bitrates.

**Strengths:**

- The paper introduces a novel ROI-guided Point Cloud Geometry Compression method. By integrating detection loss into the compression task, the method enables joint optimization of compression and detection, leading to improved detection accuracy.
- The paper introduces modules like the MSFEM and SAM to enhance the extraction of semantic information during the encoding and decoding transformation, contributing to better compression and detection results.

**Limitations:**

I apologize for my limited understanding of this field, but I believe the essay could benefit from additional clarification.

- The research primarily concentrates on two datasets, ScanNet and SUN RGB-D, potentially restricting the applicability of the findings to a broader spectrum of point cloud data. Including additional datasets like OwLii would strengthen the validity of the conclusions.
- The experimental results indicate a significant impact of compression distortion on detection tasks, particularly evident at lower bitrates. Further exploration into how different detection models interact with compression could yield valuable insights for optimizing detection performance.
- In line 734, the statement "*RPCGC outperforms these models by -0.084*" appears perplexing, as it actually implies that RPCGC performs worse than certain methods. Additionally, enhancing the explanation of metrics would enhance clarity.

**Suitability:**

3

---

### Official Review · Reviewer_E89u · 2024-05-11

**Rating:** 2
**Confidence:** 3

**Summary:**

The paper proposes a novel Region of Interest (ROI)-guided point cloud geometric compression method (RPCGC), aimed at improving the efficiency of point cloud data compression and the accuracy of object detection in machine vision tasks. Its advantage lies in achieving a high compression ratio while significantly improving the detection accuracy of point clouds, especially at higher bit rates.

**Strengths:**

see the summary part.

**Limitations:**

1.The reviewer believes that the authors' description of the model is not detailed enough. For instance, what are the specific algorithms used for quantization and de-quantization? Additionally, the authors should conduct more detailed ablation studies on each module to explore its intrinsic role. For example, how does the precision of quantization and the strategy of de-quantization affect the final compression performance and detection accuracy? How does the prediction accuracy of the ROI Prediction Network (RPN) impact compression and detection performance? The reviewer suggests that the authors add experiments to elucidate the internal workings of the model.
2. The reviewer suggests that the author's model shows limited improvement in performance compared to existing models, and the experiments conducted are insufficient. It is recommended that the authors include comparisons on more universally recognized datasets for point cloud compression, such as 8iVFB and Owlii, and also compare against more recent and state-of-the-art models.
Rui Song, Chunyang Fu, Shan Liu, Ge Li; “Efficient Hierarchical Entropy Model for Learned Point Cloud Compression”  CVPR, 2023, pp. 14368-14377
Junteng Zhang et al. “YOGA: Yet Another Geometry-based Point Cloud Compressor”, ACM MM 2023
3. The loss function of this paper includes three parts, and the reviewer believes that the authors should provide a detailed explanation of how to balance these three components. Furthermore, the reviewer is interested in the intrinsic impact of the selection of hyperparameters, such as α β γ δ, on the model. It would be beneficial to elucidate how these hyperparameters influence the model's behavior and its performance.
4. The model presented in the paper involves numerous neural network components, which suggests a potentially significant computational demand. However, the authors have not elaborated on the specifics of the deployment environment, neither in terms of hardware nor software. Additionally, there is a lack of information regarding the training overhead and the computational expenses involved in the encoding and decoding processes. The reviewer recommends that the authors provide a comprehensive account of the deployment infrastructure, including the type of hardware used, the software framework, and any optimizations that were applied. Furthermore, the authors should detail the training costs in terms of time and resources, as well as the runtime complexity for both encoding and decoding operations.

**Suitability:**

3

---

### Official Review · Reviewer_ckf1 · 2024-05-22

**Rating:** 3
**Confidence:** 2

**Summary:**

This paper presents a model for joint point cloud compression and detection, aimed at managing point cloud compression while ensuring that the reconstructed point cloud can effectively perform downstream detection tasks. Specifically, the model has two branches: one handles the encoding and decoding of coarse point clouds, and the other supplements geometric details on this basis. Experiments show that the model not only performs well in the compression task but also achieves more accurate point cloud detection results at high bitrates than traditional and learning-based compression methods.

**Strengths:**

a) The motivation is clear and important. Most point cloud compression methods focus on reconstruction fidelity, often neglecting the relationship between the reconstructed point clouds and downstream tasks. This paper proposes a multi-task learning approach to achieve simultaneous optimization for both compression and detection.

b) The proposed method is reasonable. The authors choose a joint compression and detection task, and focus on the detection part since it's more difficult.To obtain detailed results, the paper introduce a dual-branch structure, similar to global-to-local or coarse-to-fine based methods. In the enhancement layer (EL), the authors design a delicate network to capture geometry details. Also, the EL contains the ROI Prediction Network and the ROI Searching Network, making the model better suited for detection task.

c) Experiment results show that the proposed method can achieve comparable compression performance, while maintaining better detection results at high bitrates.

**Limitations:**

a) Question about contributions. In Section 2.3, the authors mentioned some similar methods in the field of joint learning of semantic information and point cloud compression, so is this task not a newly proposed one? If it is not a newly proposed task, then why are there no comparisons with this methods in the experiments? Also, the author mentioned "Although the above methods involve machine perception, most of them are simplistic and do not achieve joint optimization or supervision", I believe that if the authors could provide a more detailed explanation of the differences and contributions of this method, it would enhance readability.

b) The experiments might appear somewhat insufficient. The authors have designed a very intricate neural network, yet it seems that a considerable amount of computation is dedicated to the EL branch, which is also reasonable given that the detection task is more challenging than the compression task. However, this also leaves concern about the model's computational cost, such as the number of parameters and FLOPs. If the authors could provide a comparison of the computational costs of all methods, as well as a detailed analysis of the computational costs for each branch of the RPCGC, the paper would be more understandable. Additionally, I am interested in the results of ablation studies on some of the main modules, such as the direct removal of the EL, which the authors do not seem to have demonstrated.

c) Some equations are confusing. What is $x_{rm}$ in Equation 1, and how is $x_{enh}$ obtained? The paper mentions two loss functions, Equations 6 and 10, which are confusing regarding their relationship and differences. Also, what is the mathematical expression for $L_{decoder}$ in Equation 9?

d) Some improvements could be made to the figures. In Figure 1, it is not very clear where the model is divided into two branches. In Figure 2, it is difficult to identify which part each sub-network belongs to. In Figure 4, it would be better to include the ground truth results.

I am willing to raise my score if the author addresses the aforementioned issues.

**Suitability:**

2

---

### Official Review · Reviewer_BtPd · 2024-05-24

**Rating:** 4
**Confidence:** 3

**Summary:**

The huge data volume of point clouds poses challenges to point cloud storage and transmission, necessitating the design of point cloud compression algorithms. Current point cloud compression methods mainly focus on fidelity optimization but the practical applications require the facilitation for machine analysis. To solve this problem, this paper proposes a compression paradigm for scene point clouds like autonomous driving scenarios. The proposed framework is in dual-branch architecture, including base and enhancement layers. The base layer encodes and decodes a simplified version of point clouds, and the enhancement layer refines it by focusing on geometry details. The enhancement layer achieves this by refining the residual information through an ROI prediction network, which generates maks information and provides weighted supervision.

**Strengths:**

1) The proposed method seems novel, and the motivation of using ROI to guide the compression of scene point clouds is reasonable.
2) The experimental results are abundant to make it convincing, including the RD, R-mAP comparison, visualization, and adequate ablations. And the experimental settings are in details.

**Limitations:**

1) The paper writing is not clear and needs improvement. For example, the working mechanism of the proposed modules are described in contributions part in Introduction. I recommend the last paragraph in Introduction describes the working mechanism to make this part more readable. And the MS-FEM and SAM modules are not mentioned in the last paragraph.
2) Too many modules are mentioned in Methodology part, making it somewhat hard to read. For example, the FAM module can be confused with RAM, SAM modules, I recommend to call it Alignment module to make it clearer.

**Suitability:**

3

---

### Meta-Review · Area_Chair_7AZ5 · 2024-06-27

**Recommendation:** Accept (Poster)
**Confidence:** 5

**Metareview:**

All reviewers like to vote for acceptance of this submission. Authors need to carefully consider reviewers comments like method description, relatively poor writting, insufficient expeirments and to include them.